# Towards Consistent Video Editing with Text-to-Image Diffusion Models

**Zicheng Zhang**[1]* **Bonan Li**[1]* **Xuecheng Nie**[2] **Congying Han**[1]† **Tiande Guo**[1] **Luoqi Liu**[2]

[1]University of Chinese Academy of Sciences  [2]MT Lab, Meitu Inc.

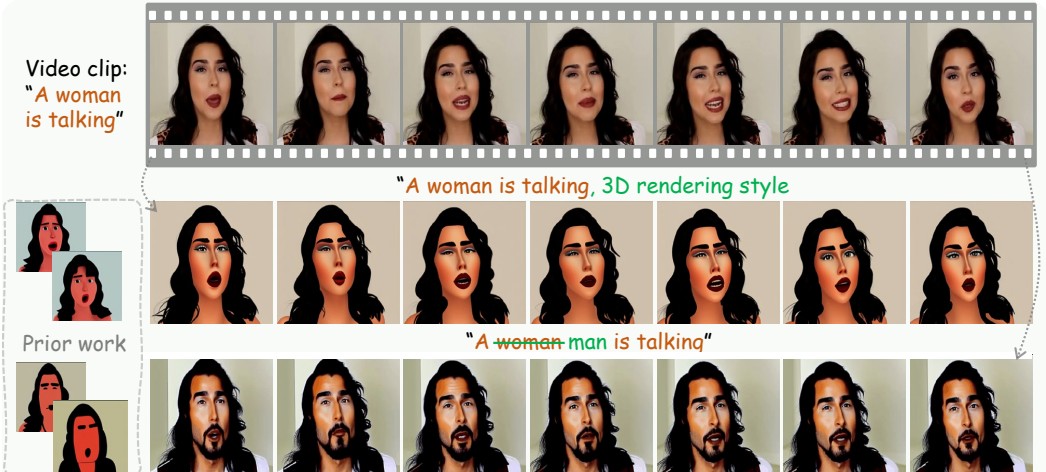

Figure 1: Comparisons of the proposed $EI^2$ and prior SOTA for text-driven video editing with Text-to-Image diffusion model. $EI^2$ addresses problems on temporal and semantic inconsistencies existing in prior work, leading to more consistent editing results. Better visual effect in color mode.

## Abstract

Existing works have advanced Text-to-Image (TTI) diffusion models for video editing in a one-shot learning manner. Despite their low requirements of data and computation, these methods might produce results of unsatisfied consistency with text prompt as well as temporal sequence, limiting their applications in the real world. In this paper, we propose to address the above issues with a novel $EI^2$ model towards **E**nhancing v**I**deo **E**diting cons**I**stency of TTI-based frameworks. Specifically, we analyze and find that the inconsistent problem is caused by newly added modules into TTI models for learning temporal information. These modules lead to covariate shift in the feature space, which harms the editing capability. Thus, we design $EI^2$ to tackle the above drawbacks with two classical modules: Shift-restricted Temporal Attention Module (STAM) and Fine-coarse Frame Attention Module (FFAM). First, through theoretical analysis, we demonstrate that covariate shift is highly related to Layer Normalization, thus STAM employs a *Instance Centering* layer replacing it to preserve the distribution of temporal features. In addition, STAM employs an attention layer with normalized mapping to transform temporal features while constraining the variance shift. As the second part, we incorporate STAM with a novel FFAM, which efficiently leverages fine-coarse spatial information of overall frames to further enhance temporal consistency. Extensive experiments demonstrate the superiority of the proposed $EI^2$ model.

---

*Equal contribution

†Corresponding author

37th Conference on Neural Information Processing Systems (NeurIPS 2023).

# 1  Introduction

In light of the surging popularity of video applications in modern times, there has been a growing focus on developing editing techniques [18, 41, 54, 45] in research community to provide automatic and efficient video editing services for counterparts, *e.g.*, content creators and media professionals. While existing techniques have explored the use of Variational Autoencoders (VAEs) [25, 32] and Generative Adversarial Networks (GANs) [12, 20, 57, 51], they are often limited to specific scenes or datasets, and struggle to provide a comprehensive solution. Thanks to the remarkable capabilities of diffusion models [15, 15, 48] in distribution learning, recent research [39, 38, 41, 36] has facilitated great progress in various fields including image synthesis, editing and restoration, posing significant promise for solving tasks in video domain. However, replicating the success in video domain is challenging, due to the higher-dimensional complexity, temporal consistency, and lack of high-quality datasets. In this paper, we develop diffusion model to tackle *text-driven single video editing* task [4, 56, 10], which utilizes text prompts to guide the change of style and objects in a given video.

From the perspective of diffusion models, prior works typically build Text-to-Video (TTV) frameworks and leverage the generative prior to address this task, wherein two primary approaches are exploited: The first approach [31, 10, 45] directly trains a TTV model on a large corpus of text and video pairs to facilitate video editing. However, due to the lack of high-quality public video datasets, current methods mainly rely on in-house datasets and demand significant computational resources. The second approach [56, 28] extends the architecture of pre-trained Text-to-Image (TTI) models [39, 38] for video editing. To capture motion in videos, various temporal modules, *e.g.*, *Spatial-Casual Attention* (*SCA*) and *Temporal Attention* (*TA*), are introduced to TTI models. The temporal motion will be learned from video via tuning and DDIM inversion. This approach is more cost-efficient, easily accessible, and thus gaining more attention in recent studies [28, 44, 35, 7, 55].

Nonetheless, methods inflating TTI models [56, 28] suffer from two critical issues: *Temporal inconsistency*, where the edited video exhibits cross-frame flickering in vision, and *Semantic disparity*, where videos are not altered in accordance with the semantics of given text prompts. Addressing these issues will considerably push the frontiers of text-driven video editing. In this paper, we reveal that the cause of the semantic disparity is not fine-tuning or over-fitting to the given video, but rather the newly introduced temporal layer, designed to ensure consistency across frames, which may weaken or even eliminate the editing power of TTI models. *We attribute this phenomenon to the covariate shift of generative feature space*, as the extra added parameters of Temporal Attention (*TA*) without any constraint inevitably transform the statistics of features [21, 52, 3]. Moreover, while using *SCA* can help reduce computational overhead, it is a suboptimal choice given that the non-autoregressive nature of the diffusion model demands considering global relationships. Therefore, incorporating efficient global spatial-temporal attention [53, 47] is essential for enhancing temporal consistency.

We present a novel approach called EI$^2$ that enhances the capabilities of existing pre-trained TTI diffusion models [39] by incorporating well-grounded temporal modules for video editing task. EI$^2$ follows the one-shot tuning paradigm [56], while making significant contributions in both theory and practice: (**1**) **Shift-restricted Temporal Attention Module** (STAM) to resolve the semantic disparity: *(i)* We provide theoretical proof, under certain assumptions, that the covariate shift is unrelated to tuning, but is caused by newly introduced parameters in the *TA* module. This provides valuable guidance in addressing the problem. *(ii)* On this basis, we identify that the *Layer Norm* [3] in *TA* module is the prime cause of covariate shift. To mitigate this issue, we propose a simple yet effective replacement called *Instance Centering* to restrict distribution shift. *(iii)* Furthermore, we constrain the shift of variance by normalizing the weights in the vanilla attention within the *TA* module. (**2**) **Fine-coarse Frame Attention Module** (FFAM) to enhance temporal consistency: We improve *SCA* with a novel fine-coarse interactive mechanism to establish an efficient spatial-temporal attention, which considers all frames while achieving low computational complexity. Instead of discarding information on the temporal dimension, we perform sampling along the spatial dimension. This preserves the overall structure of spatial-temporal data and reduces the data volumes for consideration. Concretely, FFAM respects the vanilla attention design to preserve fine information in the current frame, but downsamples non-current frames to obtain coarse features for interaction. In this way, FFAM can achieve improved modeling of motion while keeping comparable computational burden as *SCA*. (**3**) Extensive experiments validate that EI$^2$ effectively addresses semantic disparity and enhances temporal consistency in video editing compared to the current state-of-the-art methods.

## 2 Related works

**Text-to-Image diffusion models.** Diffusion models have demonstrated remarkable success in image generation by effectively approximating data distributions [15, 46, 48, 24, 9], outperforming previous generative models like GANs [6, 22, 43, 60]. Building upon the advancements of pre-trained language models [37], Text-to-Image (TTI) diffusion models [38, 41, 39] have been developed to generate high-fidelity images conditioned on text descriptions. Prominent examples include recent works like DALLE2 [38], and Imagen [41] achieve impressive and controllable image generation. Recently, the Latent Diffusion Model (LDM) [39] has drawn more attention due to its efficiency and compelling results, which is used as our base model in the experiments.

**Text-to-Video diffusion models.** TTV generation poses greater challenges compared to image generation due to its higher-dimensional complexity and the scarcity of high-quality datasets. Previous approaches mainly rely on autoregressive models [54, 19]. Pioneering works [18, 41] introduce new architectures using 3D U-Net [8] with factorized spatial-temporal attention [2, 16, 5]. Recent works attempt to leverage image diffusion priors in both modeling and data, such as Make-A-Video [45], which proposes fine-tuning from a pre-trained DALLE2 [38] with spatial-temporal modules. However, the challenges of collecting large-scale text-video datasets and the substantial computational overheads still hinder many researchers in this field. Recently, two concurrent works [23, 1] propose adapting LDM to the video without fine-tuning, but still persist inconsistent issues.

**Diffusion for text-driven editing.** Due to clear mapping relationships between data and diffusion latent space, it is efficient to obtain DDPM [15] or DDIM [46] latent of a given image and then perform text-driven editing. To strengthen structural control, P2P [13] and Plug-and-Play [50] use attention control to minimize changes to unrelated parts. Null-Text Inversion [30] fine-tunes the embedding of null text to improve real image editing. Some works [11, 40, 26] fine-tune TTI models to learn special tokens for personalized concepts and generate related images. While these methods can be applied to video editing frame by frame, it is hard for them to ensure temporal consistency. [31, 10] train TTV models to achieve impressive editing performance but are not open-source. At a earlier period, one-shot training and tuning approaches have been popular in GANs [43, 59, 58, 12] for various tasks. *Tune-A-Video* (TAV) [56] proposes the first tuning strategy on LDM [39] for video editting. Based on it, several concurrent works [28, 44, 35, 7, 55] are developed to improve the editing quality, *e.g.*, speed and background preservation. By contrast, we are concerned about the essential problem, *i.e.*, the negative impact of introduced parameters on text-driven editing ability.

## 3 Preliminaries

**Diffusion probability model.** DDPM [15] establishes a relationship between complex data distribution and the Gaussian distribution using forward and reverse Markov chains. Following the convention, we denote the random variable of data as $x_0$, and the forward process generates latent variables $x_1, \ldots, x_T$ through $q(x_t|x_{t-1}) = N(x_t; \sqrt{\alpha_t}x_{t-1}, (1-\alpha_t)I)$, where $\{\alpha_t\}_{t=1}^T$ is a fixed variance schedule. The reverse process starts from $x_T$ to sample the real data $x_0$ sequentially through

$$p_\theta(x_{t-1}|x_t, \mathcal{P}) = N(y_{t-1}; \mu_\theta(x_t, t, \mathcal{P}), \Sigma_\theta(x_t, t, \mathcal{P})). \tag{1}$$

Herein, $\mathcal{P}$ is an optional condition, $\Sigma_\theta$ can be either predefined constants or trainable parameters associated with the variance schedule, and $\mu_\theta(x_t, t, \mathcal{P}) = \frac{1}{\sqrt{\bar{\alpha}_t}}(x_t - \frac{1-\alpha_t}{\sqrt{1-\bar{\alpha}_t}}\epsilon_\theta(x_t, t, \mathcal{P}))$, where $\bar{\alpha}_t = \prod_{i=1}^t \alpha_i$, and $\varepsilon_\theta$ is a parameterized function that is trained under diffusion loss supervision

$$Loss(\varepsilon_\theta) = E_{x_0 \sim p_{data}, \varepsilon \sim N(0,I), t \sim \text{Uniform}(1,T)} \|\varepsilon - \varepsilon_\theta(x_t, t, \mathcal{P})\|_2^2. \tag{2}$$

This makes the reverse process Eq. (1) approach to the Evidence Lower Bound of data distribution.

**Latent Diffusion Model (LDM).** LDM [39] first trains an Autoencoder network consisting of an encoder $\mathcal{E}$ and a decoder $\mathcal{D}$, where the composition $\mathcal{D} \circ \mathcal{E}$ approximates the identity mapping. This enables $\mathcal{E}$ to compress image data into a low-dimensional latent space. LDM then learns a diffusion model by optimizing Objective (2) in the latent space, with the corresponding text prompt $\mathcal{P}$. In LDM, $\epsilon_\theta$ is equipped with a convolutional Transformer-based U-Net architecture, where the attention mechanism plays a critical role in interacting information from different modalities. Specifically, given two features $z \in \mathbb{R}^{l_1 \times d_1}$ and $c \in \mathbb{R}^{l_2 \times d_2}$, the attention mechanism [53] obtains three elements Query $Q$, Key $K$ and Value $V$ by $Q = zW_q$, $K = cW_k$, $V = cW_v$, where $W_q \in \mathbb{R}^{d_1 \times d}$, $W_k$ and $W_v \in \mathbb{R}^{d_2 \times d}$. After that, they will interact to generate the transferred feature via

$$\text{Attention}(z, c) = M_{Q,K}V, \text{ where } M_{Q,K} = \text{softmax}(QK^T/\sqrt{d}). \tag{3}$$

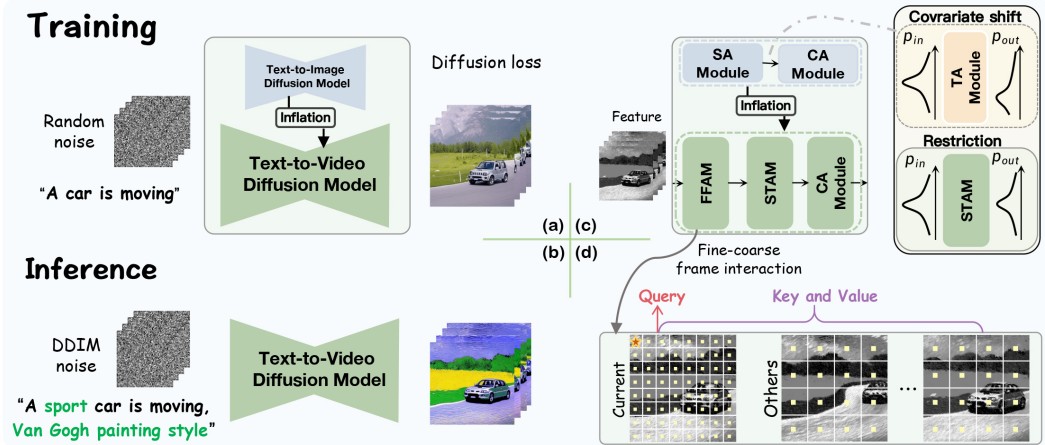

Figure 2: Illustration of EI$^2$ for text-driven video editing. (a) One-shot tuning paradigm: Given a video and a text prompt, EI$^2$ first inflates a pre-trained TTI model into a TTV model, which is further tuned by minimizing the diffusion loss. (b) Video editing: EI$^2$ uses noise from DDIM inversion [15] and a custom text prompt to generate the edited video. (c) Model details: EI$^2$ enhances the TTI model by upgrading its Self Attention (*SA*) module to a Fine-coarse Frame Attention Module (FFAM) and introducing a novel Shift-restricted Temporal Attention Module (STAM). Unlike Temporal Attention (TA) Module which shifts the input distribution intensively, STAM constrains the output to better align with the subsequent modules. (d) Interactive mechanism in FFAM: For each patch (red star) in the latent, it interacts with the fine patches in the current frame and coarse patches in other frames.

Let define $norm$ as the *Layer Norm* operator [3] and $Linear(z) = zW_L + b$, where $W_L \in \mathbb{R}^{d \times d_1}$ and $b \in \mathbb{R}^{d_1}$, a Transformer block in $\varepsilon_\theta$ has the residual structure shown below

$$\text{Transformer}(z, c) = z + Linear(\text{Attention}(norm(z), norm(c))). \quad (4)$$

For convenience in later discussions, we include the normalization operation of $c$ in the equation, although it may not be implemented in practice. LDM incorporates two types of Transformer blocks. The first Self Attention module only receives $z$ to compute $\text{Transformer}(z, z)$, and the other Cross Attention module takes the latent feature $z$ and the text embedding $c$ to compute Eq. (4). We also overlook the Dropout [49] in the transformer block, as it freezes during the fine-tuning process.

**Notations.**   For clarity, we abbreviate Self Attention, Cross Attention, and Temporal Attention as *SA*, *CA*, and *TA*, respectively. Accordingly, the Transformer blocks in Eq. (4) associated with these attention mechanisms are named *SA* Module, *CA* Module, and *TA* Module. In this paper, we use variables $n$, $l$ and $d$ to represent the length of temporal, spatial, and feature dimensions, respectively.

## 4   Methodology

### 4.1   Overview of EI$^2$

Given a video clip $\mathcal{V} = \{v^1, \ldots, v^n\}$ consisting of $n$ frames, and an associated text prompt $\mathcal{P}$, text-driven video editing task leverages other text prompts to guide style or object changes in the video. While existing large-scale TTI diffusion models [39, 38, 41] have shown exceptional behavior in image editing, they suffer from temporal inconsistency when applied to the video frames.

**One-shot tuning for video editing.**   In this paper, we propose a novel approach, named EI$^2$, which inflates the frequent pre-trained TTI diffusion model LDM [39] with well-designed temporal modules to create a TTV diffusion model. As depicted in Figure 2, the inflated model is optimized to minimize diffusion loss about data pair $(\mathcal{V}, \mathcal{P})$, so as to capture the temporal motion in video $\mathcal{V}$, while retaining the editing capability of TTI model. To edit the video, we use an altered prompt to guide the diffusion reverse process based on Eq. (1), where the initial latent $x_T$ is obtained by the DDIM inversion [15].

**Model Inflation.**   Technically, during the inflation process, the initial step involves converting all spatial convolutions in LDM into pseudo 3D convolutions, which allows video data and latent to be processed in the same way as a batch of images. Furthermore, our approach differs from

previous work [56, 15, 35, 28] in two key aspects within Transformer blocks: *(a)* We introduce a novel Shift-restricted Temporal Attention Module (STAM) that learns motion while solving the covariate shift problem. This addresses the essential limitation in previous works [56, 28], where the text-driven capability of the TTI model is weakened or deprived after inflation with a *TA* Module. *(b)* Recognizing the complexities over time, we upgrade the existing *SA* Module into a new Fine-coarse Frame Attention Module (FFAM). Unlike *SCA* in previous approaches [56, 28] reduces the complexity of attention through causal sampling along the time dimension, our FFAM compresses information along spatial dimension while retaining temporal information to facilitate efficient and effective spatial-temporal attention interaction, leading to better results.

## 4.2 Theoretical analysis of covariate shift problem

As depicted in Figure 1, it is apparent that previous works [56, 28] can lead to semantic disparity, *i.e.*, inconsistencies between the text prompt and the edited video. We attribute this disparity to the covariate shift [21] in the feature space, where modifying the prior distribution of a pre-trained generative model typically leads to a degradation in synthesis. In the case of *Tune-A-Video* [56], the one-shot data poses difficulties in training meaningful parameters for aligning the output distribution from *TA* Module with the demands of the subsequent module. Therefore, our analysis investigates the transition of feature distribution in TTV model and provides evidence of covariate shift.

Since *Tune-A-Video* solely tunes the parameters of Transformers including $W_q$ in the vanilla modules, and all weights in *TA* Module, we abstract the problem as changes in distribution after passing through these weights. By considering the structures in Eq. (3) and Eq. (4), which serve as the basis for the *SA*, *CA* and *TA* Modules, we can obtain the following discoveries:

**Proposition 1.** *We assume that for Transformers in Eq. (4), any input feature $z \in \mathbb{R}^{l \times d}$ (or $\mathbb{R}^{n \times d}$) is a sample of i.i.d $d$-dimensional Gaussian random variables, denoted as $rv(z) \sim N(\mu_{rv(z)}, \Sigma_{rv(z)})$, and $\hat{z} = Norm(z)$, same notation for c. (i) For the output from Eq. (3), the $i$-th row vector variable, $\mathrm{Attention}(z, c)^i$, follows the Gaussian distribution $N(\mu_{rv(V)}, \omega_{Q,K}\Sigma_{rv(V)})$, where $\mu_{rv(V)} = W_v^T \mu_{rv(\hat{c})}$, $\Sigma_{rv(V)} = W_v^T \Sigma_{rv(\hat{c})} W_v$, and $\omega_{Q,K} = \|M_{Q,K}^i\|_2^2$. (ii) For the output from Eq. (4), each row vector variable in $\mathrm{Transformer}(z, c)$ follows the Gaussian distribution $N(\mu'_{rv(z)}, \Sigma'_{rv(z)})$, where $\mu'_{rv(z)} = \mu_{rv(z)} + W_L^T \mu_{rv(V)} + b$, and $\Sigma'_{rv(z)} = \Sigma_{rv(z)} + \omega_{Q,K} W_L^T \Sigma_{rv(V)} W_L$.*

The proof is provided in Supplementary Materials. We remark that *(i)* The Gaussian assumption is common and basic in modern neural networks [21, 3, 52], as it is efficient to characterize feature distributions. Here we leverage its properties, particularly additivity, to analyze changes in distributions. *(ii)* When tuning the weight matrix $W_q$ in *SA* and *CA*, the mean $\mu'_{rv(z)}$ remains unaffected, while the covariance $\Sigma'_{rv(z)}$ is primarily influenced by the modification of $\omega_{Q,K}$. However, since $W_q$ is well initialized from the pre-trained TTI model, its impact on the covariate shift is considered negligible. *(iii)* The weights in *TA* Module, which are randomly initialized and learned from one-shot data, have the potential to greatly influence the feature distribution, whereas it is crucial to ensure the output distribution resembles the input distribution to properly fit the following pre-trained modules.

## 4.3 Shift-restricted Temporal Attention Module

Based on the analysis, we can explicitly identify that *TA* Module leads to covariate shift. To address this issue, we propose to improve this module by restricting the distribution shift. In concrete, given that each unit in the spatial grid has temporal feature $z \in \mathbb{R}^{n \times d}$, when passing *TA* Module it will be transformed by the operation $\mathrm{Transformer}(z, z)$ in Eq. (4). Therefore, our main objective is to minimize the diffusion loss while ensuring *TA* Modules are constrained in distribution shift, *i.e.*,

$$\min_{\theta} Loss(\varepsilon_\theta), \quad \text{where } \textit{TA} \text{ Modules subject to } \begin{cases} \mu_{rv(z)} + W_L^T \mu_{rv(V)} + b \longrightarrow \mu_{rv(z)} \\ \Sigma_{rv(z)} + \omega_{Q,K} W_L^T \Sigma_{rv(V)} W_L \longrightarrow \Sigma_{rv(z)} \end{cases} \quad (5)$$

It is worth noting that there exist trivial solutions for *TA* Modules, such as vanishing all parameters. However, we seek to identify the optimal configuration that maximizes the contribution to the task.

**Plight from *Layer Norm*.** As an essential operation, nevertheless, we find *Layer Norm* [3] in the *TA* Module significantly contributes to the issue of covariate shift. The computation of Layer Norm across all dimensions, *e.g.*, the mean value $\mu_z = \frac{1}{nd}\sum_{i,j=1}^{n,d} z^{i,j}$, fails to properly center the

feature along the temporal dimension. This results in non-zero values for $\mu_{rv(\hat{z})}$ and $\mu_{rv(V)}$, leading to covariate shift and conflicts with the objective (5). Experimental results demonstrate a notable discrepancy between $\mu_{rv(z)}$ and $\mu_z$, with the former ranging from $1e-1$ to 1, while the latter is typically less than $1e-2$. Therefore, we can conclude that *Layer Norm inadequately centers the feature in the temporal dimension, leading to a shift in the mean value.* This conclusion is supported not only by theoretical analysis but also by compelling empirical evidence (see Section 5.2 and Figure 4), where even minor refinements in *Layer Norm* can yield noticeable improvements in the results.

**Instance Centering layer.**    To address the issue of *Layer Norm* in *TA* Module, we propose *Instance Centering* ($IC$) as an effective replacement, which operates on temporal data $z \in \mathbb{R}^{n \times d}$ by

$$IC(z) = z - \frac{1}{n}\Sigma_{i=1}^{n}z^i. \tag{6}$$

When applied to a batch of temporal data $z \in \mathbb{R}^{l \times n \times d}$, it is defined to individually operate on each sample. Consequently, the $IC$ layer guarantees that $\mu_{rv(\hat{z})}$, $\mu_{rv(V)}$, and $W_L^T \mu_{rv(V)}$ in Objective (5) vanish in principle. In this way, the new parameters in *TA*, except for the bias, would not affect the mean of distribution. Importantly, $IC$ layer preserves the temporal variance of the data. This deviation from the *Instance Norm* [52], which normalizes the data along the sequence dimensions, offers not only computational efficiency, resulting in about 20% increase in computational speed, but also plays a role in controlling variance shifts associated with subsequent techniques.

**Normalized linear mapping.**    Due to the interaction of multiple variables, the change of temporal variance in *TA* Module is unavoidable. We can establish the relation using norm properties:

$$\|\Sigma'_{rv(z)} - \Sigma_{rv(z)}\| \leq \|W_L\|^2 \|W_v\|^2 \|\Sigma_{rv(\hat{z})}\|. \tag{7}$$

When considering the Frobenius norm, this equation indicates that the change in each element of the covariance is controlled by the rescaling coefficient in $Norm$, matrix norm of $W_L$ in $Linear$, and $W_v$ in *TA*. Therefore, it implies that we can mitigate the variance shift by regularizing these values. From this we design the following improvements: *(a)* We do not employ the rescaling operator in $IC$ since the order of most temporal feature variances is varied from $1e-2$ to $1e-1$, meaning the operation would amplify the variance and magnifying the covariate shift. This observation is also supported by qualitative results (see Figure 4). *(b)* Instead of adding a regularization term to loss, we employ a more efficient technique called spectral normalization ($SN$) [29] to rescale $\bar{W} = W/\sigma_{\max}(W)$, where $\sigma_{\max}$ is the spectral norm of matrix. $SN$ has been widely used in various tasks and offers better stability and theoretical guarantees compared to other weight normalization [42]. In our approach, we leverage $SN$ to redefine the $\bar{\text{Attention}}$ and $\bar{\text{Linear}}$ layers as follows:

$$\bar{\text{Attention}}(z, c) = M_{Q,K}z\bar{W}_v, \text{ and } \bar{Linear}(z) = z\bar{W}_L + b. \tag{8}$$

**STAM.**    The STAM layer is an extension of Transformer in Eq. (4) and formulated as follows:

$$\text{STAM}(z) = z + \bar{Linear}(\bar{\text{Attention}}(IC(z), IC(z))). \tag{9}$$

We can prove that the feature vector in $\text{STAM}(z)$ follows a distribution $N(\mu_{rv(z)} + b, \Sigma'_{rv(z)})$, where $\|\Sigma'_{rv(z)}\| \leq 2\|\Sigma_{rv(z)}\|$ under the spectral norm. Since the bias $b$ is initialized from zero, the impact on $\mu_{rv(z)}$ is negligible under a small learning rate. Therefore, this design effectively alleviates the covariate shift in theory, enabling the improvement of semantic disparity in practice.

### 4.4 Fine-coarse Frame Attention Module

Considering the set $\mathcal{Z} = \{z^j \in \mathbb{R}^{l \times d}\}_{j=1}^{n}$, which comprises features of all frames, *Tune-A-Video* introduces a Sparse-Casual Attention (*SCA*) mechanism to enhance the *SA* with sparse-temporal interaction, where only the first frame and its immediate previous frame are used for interaction, instead of considering all frames. This significantly reduces the computational complexity from $\mathcal{O}(n^2l^2)$ to $\mathcal{O}(2nl^2)$. While *SCA* is more effective than SA, it is not optimal for video editing tasks, in which global spatial-temporal relationships are crucial to ensure overall coherence. To address this issue, we propose the Fine-coarse Frame Attention Module (FFAM) as a new solution. FFAM leverages a fine-coarse strategy to incorporate fine-grained information from the current frame with coarse features from other frames. Specifically, FFAM is defined as follows:

$$\text{FFAM}(z^i; \mathcal{Z}) = \text{Attention}(z^i, z_{FC}),$$
$$\text{where } z_{FC} = \text{concat}[z^i, \{\text{downsample}(z^j, r)\}_{j \neq i}] \in \mathbb{R}^{(\frac{n-1+r^2}{r^2}l) \times d}, \tag{10}$$

Table 1: Quantitative comparison with evaluated baselines. The "Training" refers to the process of optimization and DDIM inversion [15], and "Memory" refers to the peak footprint of GPU device.

| Method | Frame consistency | | Textual alignment | | Runtime [min] | | Memory [Gib] | |
|---|---|---|---|---|---|---|---|---|
| | CLIP Score↑ | User Vote↑ | CLIP Score↑ | User Vote↑ | Training↓ | Inference↓ | Training↓ | Inference↓ |
| *Tune-A-Video* [56] | 96.64 | 25.6% | 28.72 | 15.4% | 11.0 | **0.5** | **9.6** | **5.3** |
| *Video-P2P* [28] | 96.84 | 28.3% | 28.24 | 19.5% | 19.2 | 2.0 | 27.3 | 19.5 |
| *Vid2Vide-zero* [33] | 95.17 | 11.2% | 29.39 | 22.4% | **2.5** | 2.5 | 17.2 | 23.3 |
| $EI^2$ | 95.94 | **34.9%** | **29.84** | **42.7%** | 12.5 | **0.5** | 11.0 | **5.3** |

where $\text{downsample}(\cdot, r)$ rescales the input with ratio $1/r^2$. The design of FFAM respects the rule of *SA* to leverage data in the current frame, and uses coarse features sampled from other frames as an aid. In contrast to *SCA*, which samples data over time to improve efficiency, FFAM adopts a robust way that condenses data along the spatial dimension, while retaining the structural relationships in both spatial and temporal dimensions. It is worth noting that FFAM reduces the complexity to $\mathcal{O}(\frac{n-1+r^2}{r^2}nl^2)$. In experiments, when $r$ is set to 2 or 4, FFAM obtains improved temporal consistency while achieving comparable speed to *SCA*. This trade-off between *SA* and global spatial-temporal interaction enables FFAM to strike a balance between efficiency and consistency in video tasks.

## 5 Experiments

**Implementation details.** Our implementation of $EI^2$ is based on the stable diffusion v1-4 framework[3]. We keep the Autoencoder of LDM frozen and tune $W_Q$ of the FFAM and *CA* Modules, and all parameters of STAMs. We follow previous works [56] to perform tuning on 8-frame videos of size $512 \times 512$. We utilize the AdamW optimizer with a learning rate of $3e - 5$ for a total of $500$ steps. During inference, we initialize the model from the DDIM inversion[46] and set the default classifier-free guidance [17] to 7.5. All experiments are conducted on an NVIDIA Tesla V100 GPU.

**Baselines.** In view of the limited accessible resource in this field, we select the three representative methods as baselines. *(1) Tune-A-Video* [56]: The current SOTA in the field and the conventional baseline for related works. *(2) Vid2Vid-zero* [55]: A TTI-based zero-shot video editing method without fine-tuning. *(3) Video-P2P* [28]: An improvement method over *Tune-A-Video* via using P2P [13] and Null-Text [30]. Experiments are conducted with their official codes and configurations.

**Datasets.** Following previous works [56, 28], we collect videos from the DAVIS dataset [34] for comparison. We also gather face videos from the Pexels website to assess the fine-grained editing in the face domain. We utilize a captioning model [27] to automatically generate the text prompts.

### 5.1 Comparison

**Qualitative results.** We present qualitative comparison results between $EI^2$ and baselines in Figure 3. Due to limited space, additional results are provided in the *Supplementary Materials*. We evaluate the performance of the methods using different general scenarios including object and style editing. The following conclusions can be drawn from the results: *(1) Tune-A-Video* [56] tends to performs better in coarse-level object replacement but poorly in terms of style. This is evident in its attempt to forcefully fit the "surfing" action for object replacement, resulting in noticeable artifacts in the generated results. It also fails to produce the desired stylistic effects and still maintains the photorealistic style of the original videos. *(2) Video-P2P* [28], based on *Tune-A-Video*, improves consistency with the original video through attention control but struggles with global style as before. *(3) Vid2Vid-zero* [55] does not use the *TA* Module and adopts the parameters of the source model entirely, avoiding the issue of covariate shift. However, the lacking of fine-tuning procedure leads to the edited results could not faithfully preserve the characteristics of source videos. *(4)* Compared to previous works, our approach combines the fidelity to the original video from *Tune-A-Video* with the editing power of the TTI model from *Vid2Vid-zero*. For instance, in the case of "surfing", $EI^2$ effectively preserves the motion of source videos while creating frames suitable for characters, instead of blindly copying the source motion and introducing artifacts. Our experiments demonstrate that $EI^2$ exhibits significant visual advantages over the competitors.

**Quantitative results.** In line with previous studies [56, 28, 55], we evaluate methods using CLIP score [14] and user study to assess frame consistency and textual alignment. *(1) CLIP score*: To

---

[3]Stable Diffusion: `https://huggingface.co/CompVis/stable-diffusion-v1-4`

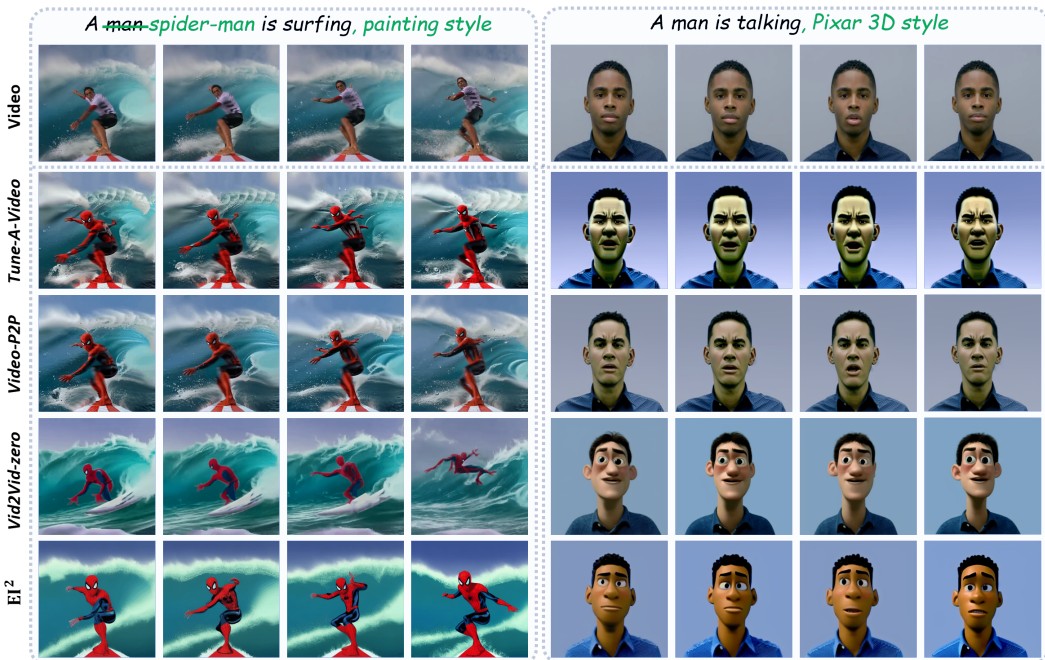

Figure 3: Comparison of EI$^2$ with *Tune-A-Video* [56], *Video-P2P* [28], and *Vid2Vid-zero* [55]. The black text in the first line represents the video caption, while the green text indicates the prompt for revision. The outputs generated by EI$^2$ exhibit greater temporal continuity and semantic consistency.

evaluate frame consistency, we calculate CLIP [37] image embedding for all frames in the edited videos and report the average cosine similarity between pairs of video frames. For measuring textual faithfulness, we compute the average CLIP score between frames of output videos and corresponding edited prompts. We employ 15 videos from the dataset and edit them in terms of object and style, resulting in a total of 75 edited videos for each model. Table 1 details the average results, which indicate that our method exhibits the strongest ability to achieve semantic alignment. Nevertheless, it is important to note that the automatic evaluation may not align perfectly with human perception [31], thus should only serve as reference scores. *(2) User study*: We present 30 pairs of videos and text prompts to volunteers and ask them to vote for edited videos with the best temporal consistency and those that best match the textual description. We collect valid votes from 50 volunteers. Table 1 illustrates that EI$^2$ receives the highest number of votes in both aspects, indicating that our method achieves superior editing quality and is preferred by users in practice.

**Resource consumption.** The practical resource consumption of video editing is a significant consideration. The results in Table 1 show that EI$^2$ introduces only a marginal increase in computational overhead compared to *Tune-A-Video*, while achieving a substantial improvement in performance. This indicates that our approach effectively balances the trade-off between resource utilization and editing performance, making it a practical and efficient solution for video editing task.

## 5.2 Ablation Study

**Restricting covariate shift.** As the core of our research, we evaluate the impact of the covariate shift problem on video editing performance. According to the theoretic analysis in Sec. 4.2, we focus on the *Layer Norm* within the *TA* Module of *Tune-A-Video* and make modifications while keeping the rest of the architecture unchanged. These modifications are performed based on the official code of *Tune-A-Video*. Figure 4 presents the results of these experiments. In the second column, it is observed that the vanilla *Tune-A-Video* tends to restrain the textual guidance, leading to inadequate editing performance. Removing the rescaling parameters of the *Layer Norm* or completely eliminating the module itself does not effectively alleviate the covariate shift problem. However, when we replace the module with the *Instance Norm* [52], a notable improvement in performance is achieved. Moreover, incorporating *IC* and weight normalization further enhance the results. These findings indicate a significant relationship between covariate offset and semantic disparity, affirming the effectiveness of our approach in mitigating these challenges.

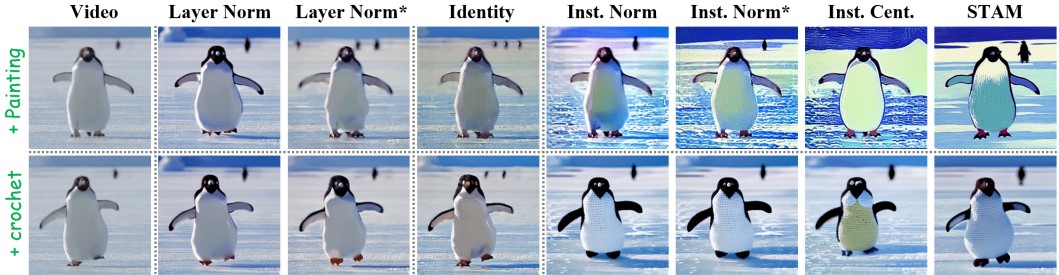

Figure 4: Ablation on the restriction of covariate shift. "**\***" means that the operator does not use learnable weights. Replacing *Layer Norm* with instance-based operator can obviously mitigate the covariate shift and has a very distinct effect on improving textual guidance.

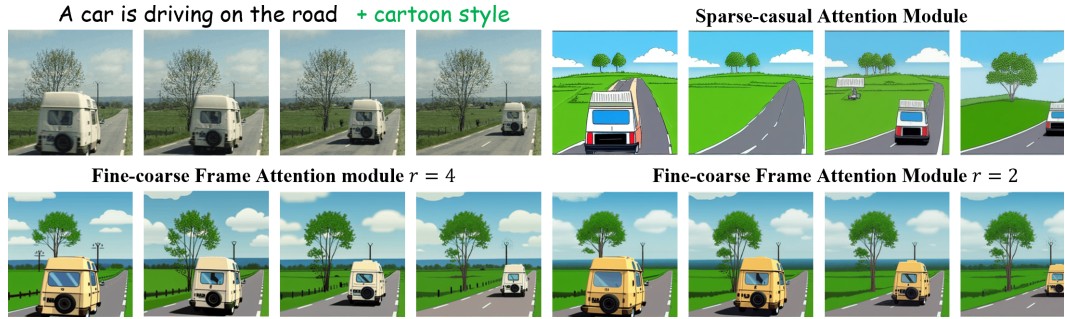

Figure 5: Ablation on the spatial-temporal attention mechanism. Compared with previous *SCA* Module [56], our FFAM can enhance the temporal consistency even with a large scale ratio.

**Fine-coarse Frame Attention.** Figure 5 illustrates the effectiveness of the fine-coarse interaction in improving temporal consistency. When combining the *SCA* Module with STAM, it cannot effectively learn temporal information, resulting in obvious flickering. In contrast, FFAM effectively addresses this issue, ensuring superior editing quality and temporal coherence in our model. It is important to note that the scale ratio used in FFAM needs to be within the range of the LDM feature map, which ranges from 8 to 64. Moreover, applying full spatial-temporal attention to 8 frames requires approximately 32GiB memory footprint and a significant amount of time (∼45min) for training, which is not practical in application. In this regard, our FFAM provides a flexible alternative to spatial-temporal interaction, addressing the limitations associated with resource consumption while still achieving satisfactory results in terms of temporal consistency.

## 6 Conclusion

In this paper, we propose the $EI^2$ model to tackle inconsistent editing results in expanding TTI models for video editing. Our analysis reveals that semantic inconsistency arises from new parameters in temporal attention, leading to covariate shift and compromising editing capability. The $EI^2$ model comprises two key modules: the Shift-restricted Temporal Attention module (STAM) and the Fine-coarse Frame Attention module (FFAM). STAM employs a simplified Instance Centering layer to eliminate mean shift and normalized attention mapping to constrain variance shift. FFAM, integrated with STAM, effectively utilizes fine-coarse spatial information across frames to enhance temporal consistency. Extensive experiments confirm the validity and effectiveness of our approach. We hope our work provides insights for future research in related fields.

**Limitations.** Our work has two main limitations. Firstly, while we believe that our analysis and proof of the covariate shift provides valuable insights into related issues, it heavily relies on the Gaussian assumption, which cannot hold exactly in practical settings. Secondly, although our method demonstrates stronger editing capabilities, it still suffers from failures of temporal consistency in some cases, such as when replacing the "surfing man" with animals. We observe that $EI^2$ performs better in most style editing, but the replacement for objects is limited to similar attributes. Further investigation and improvement are necessary to address these limitations.

## Acknowledgements

This paper is supported by the Strategic Priority Research Program of Chinese Academy of Sciences (XDA27000000), the National key research and development program of China (2021YFA1000403), the National Natural Science Foundation of China (Nos. U19B2040, 11991022) and the Fundamental Research Funds for the Central Universities.

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
