# OpenReview forum: "Towards Consistent Video Editing with Text-to-Image Diffusion Models"
_NeurIPS.cc/2023/Conference — NeurIPS 2023 poster_

### Official Review · Reviewer_ipyC · 2023-06-30

**Soundness:** 2 fair
**Presentation:** 3 good
**Contribution:** 2 fair
**Rating:** 4
**Confidence:** 5

**Summary:**

Previous works on one-shot video editing might produce results of unsatisfied consistency with text prompt as well as temporal sequence. This paper presents that such inconsistency is caused by covariate shift in the feature space. Therefore, they propose Shift-restricted Temporal Attention Module (STAM) and Fine-coarse Frame Attention Module (FFAM) to improve the performance.

**Strengths:**

1 The theoretical analysis of covariate shift is comprehensive and clear.

2 The proposed STAM can well address the style change problem while FFAM can improve the temporal consistency.

**Weaknesses:**

1 The motivation for STAM is that the inconsistency between text prompt and video lies in the covariate shift between the pretrained T2I model and the fine-tuned T2V model. This paper provides a theoretical analysis of this. However, as the output layer of temporal attn is initialized as zeros and fine-tuned by a small learning rate, the influence of covariate shift may not be much significant. It is better to analyze the covariance quantitatively.

2 To address the covariance shift, in addition to the IC layer, spectral normalization is also proposed. But this paper does not provide an explanation and analysis of spectral normalization.

3 **Frame Consistency**: Fitting to the one-shot video improves the video consistency while STAM aligns the feature to the T2I distribution. Therefore, the two parts are contradictory, and the frame consistency is degraded by this. This is a severe limitation of this paper. In Table 1 and Figure 3, the temporal consistency of this paper is much worse than Tune-A-Video and Video-P2P.

4 The applications are limited. The proposed method mainly focuses on the style transfer problem. Other editing applications, such as background change, are not considered and compared with previous works.

5 Despite providing ablation videos in the supplementary material, the ablation studies in the main paper are not comprehensive. More video frames should be provided to show the influence on frame consistency.


**Questions:**

My main concern is the frame consistency problem. The quantitative analysis of covariate shift and the explanation of spectral normalization are also important.

**Limitations:**

I think the frame consistency problem is a limitation that is ignored by the authors.

---

> ### Author Rebuttal · Authors · 2023-08-09
>
> __Q1__: The influence of covariate shift may be small. It is better to analyze the issue quantitatively.
>
> __A1__: Thanks for pointing out. We agree with your point that the implementations have potentially alleviated the covariate shift. But, due to the resonance of multiple variables, the issue is still present and significant.
> - Theoretical analysis. Prop. 1 shows $W_{L}$ and $\mu_{rv(V)}$ jointly determine mean and value, making it essential to consider both their impacts. Moreover, the weight of output layer has a dimension $h^2$, where $h=320,640,1280$. Hence, its norm will quickly increase from zero even with a small learning rate $lr$, _e.g._, $O(h * lr)$ for F-norm at each iteration.
>
> - Quantitative analysis. With 16 TA modules inserted to the inflated LDM, we measured the L2 difference of mean and variance before and after the TA module. As the absolute value scales vary across training videos and layers, it is hard to provide meaningful statistic for them. We report the empirical range of relative changes (_e.g._, $||\\mu_{rv(z)}'-\\mu_{rv(z)}||/||\\mu_{rv(z)}||\$) for 10 videos.
>   - For TAV: The relative change in mean reveals that the first 6 layers generally have minor shifts less than 10%. However, subsequent layers almost exhibit larger than 10%, and some even surpass 100%. Variances have similar trends, with the first 6 layers having roughly 1% changes, and subsequent layers having around 10% changes. *Given the cumulative impact of these layer-by-layer shifts*, the effect of covariate shift would be notable.
>   - For EI$^2$: The affect in mean largely decreases, with the first 6 layers changing nearly 1% and the later layers having changes less than 10%. Variance values also demonstrate about half reduction. These align with our theoretical analysis, where it is easier to control mean shift.
>
> Hence, the covariate shift indeed exists. The results in the paper further confirm this point. We will add these in the revision.
>
> __Q2__: Explanation and analysis of spectral normalization (SN).
>
> __A2__: As the Line 223-225, SN, i.e., $\bar{W} =W / \sigma_{\text{max}}(W)$, is a simple but effective way to normalize the matrix, where $\sigma_{\text{max}}(W)$ is the max singular value, or called the spectral norm. More details please refer to A2 for Reviewer k6zc.
>
> We mainly leverage SN to control the variance shift during STAM (Eq. 7). For TAV,  $\sigma(W_L)$ concentrates in $(0,2)$, while $\sigma(W_v)$  concentrates in $(3,6)$. After the square product and layer-by-layer cumulation, it will cause significant shift. Hence, SN plays a key role for controlling this no more than 1 in normalized linear mapping (A2 for Reviewer FCWG).
> We will add these in the revision.
>
> __Q3__: The contradictory between learning video consistency and aligning the feature in STAM.
>
> __A3__: We clarify that the two aspects are not contradictory but rather complementary components that contribute to achieving impressive editing results. STAM still aims to fit the motion of video, but just under extra regularization of feature alignment.
>   1. TA module is crucial for consistency but faces covariate shift issue. Given that the removal of it largely degrades temporal consistency, STAM offers a grounded approach to extend it.
>   2. Our analysis provides insight based on "certain assumption". That implies we cannot entirely eliminate, but just "alleviate" the issue. For example, while the $IC$ layer theoretically outputs random variables with a zero mean, this process is biased in practice due to limited samples.
>   3. In the above analysis, we still observe shifted mean and variance in output features, indicating that STAM captures temporal motion from the video.
>
> __Q4__. The frame consistency degraded by STAM is a severe limitation of this paper.
>
> __A4__: We recognize that  STAM may decrease the temporal consistency. Nevertheless, it is not a severe limitation for inconsistency in practice:
> 1. It is important to emphasize the goals of text-driven video editing, where both the temporal and frame consistency should be considered.
>
> 2. As shown in Table 1 of the PDF, STAM just slightly performs worse than vanilla TA, but brings considerable benefits to improve the textual alignment, thus enhance overall editing effect.
>
> 3. The degradation might not solely result from STAM but could also the limitations of one-shot tuning. Evidenced by recent methods like TAV, greater divergence between the editing result and the original video tends to worsen temporal consistency. STAM can produce more pronounced effects, potentially leading to increased inconsistency. For simply edits such as converting "car" to "sport car", STAM can achieve comparable consistency as TA.
>
> 4. The consistency can be further enhanced with other modules without degrading much semantic consistency, such as FFAM. Thus EI$^2$ strikes an improvement for semantic consistency, while still ensuring temporal consistency.
>
> __Q5__: In Table 1 and Figure 3, the temporal consistency of this paper is much worse than Tune-A-Video and Video-P2P.
>
> __A5__:  We would like to clarify the distinction between the "CLIP score for frame consistency" and conventional "temporal consistency (Please refer to A2 for Reviewer 8VBg). And we respectfully disagree that the results of EI are much worse:
>   1. Neither TAV or Video-P2P achieves the desired effect, and artifacts even appear.
>   2. The videos in SM show that our edited videos are visually continuous and natural.
>   3. A large number of user studies to illustrate the advantages of our approach.
>
> __Q6__: Validating method in more general application.
>
> __A6__: We show some examples for object, background changes in the pdf.  EI$^2$ can perform well, even better than TAV in these situations.
>
> __Q7__: Ablation studies in the main paper are not comprehensive.
>
> __A7__: We will refine these in the revision. And we provide extra quantitative ablation study in the PDF. Please refer to our response to Reviewer FCWG.

---

> > ### Comment · Reviewer_ipyC · 2023-08-17
> >
> > Thanks for your rebuttal.
> >
> > As for your response to my question about "spectral normalization", you ask me to refer to A2 for Reviewer k6zc. But "A2 for Reviewer k6zc" seems to explain a different question.

---

> > > ### Author Response · Authors · 2023-08-17
> > > **About spectral normalization**
> > >
> > > Dear reviewer ipyC
> > >
> > > Thanks for your prompt response. We apologize for any confusion caused. To accommodate the word limit, we intend to utilize the reference to offer the implementation of spectral normalization (SN) within the context of STAM.  Here we provide a comprehensive elucidation:
> > >
> > > - Formulation: SN offers a simple and straightforward way to normalize the matrix weight, denoted as $\bar{W} =W / \sigma_{\text{max}}(W)$, where $\sigma_{\text{max}}(W)$ is the max singular value, or called the spectral norm of $W$. After applying SN, we attain $\sigma_{\text{max}}(\bar{W})=1$, and $||\bar{W}||^2=1$ under the spectral norm.  Drawing from Equation 7, where $||\Sigma_{rv(z)}' - \Sigma_{rv(z)}|| \leq || W_{L}||^2 || W_{v} ||^2 ||\Sigma_{rv(\hat{z})}||$,
> > >  we primarily exploit SN to redefine linear operator within temporal attention. This will eliminate the influence of $W_{L}$ and $W_{v}$ on variance and make
> > > $||\Sigma_{rv(z)}' - \Sigma_{rv(z)}|| \leq ||\Sigma_{rv(\hat{z})}||$ in theory.
> > > - Implementation: Recognizing that computing exact singular values for high-dimensional matrices is often time-costing,  we follow established works [1] by employing the Power iteration algorithm to fast approximate the max singular value. Consequently, the integration of spectral normalization does not result in additional time overhead.  To avoid potential complications arising from small singular values during normalization, we practically apply spectral normalization solely to matrices with a spectral norm exceeding 1.
> > > - Effects: In most experiments for TAV,  a distinct trend can be observed. Specifically, due to zero initialization, $||W_{L}||$ in temporal attention tends to concentrates within $(0,2)$, while $||W_{v}||$  tends to  concentrates in $(3,6)$. Given the cumulative effect of squaring and layer-by-layer accumulation,  the pivotal role of SN becomes evident in controlling these terms, validated by the quantitative analysis in A1. Moreover, the quantitative  ablation study provided in table 1 of the PDF file indicates that, directly leveraging Normalized Linear Mapping (NLM) can obviously improve the textual alignment score (29.02) compared with baseline model (+FFAM 28.50). The qualitative results in Figure 4 also validates this point. Thus, SN does play an important role in alleviating the covariance shift.
> > >
> > > We are pleased to offer further explanation on any remaining questions you might still have.
> > >
> > > [1] Miyato, Takeru, Toshiki Kataoka, Masanori Koyama and Yuichi Yoshida. “Spectral Normalization for Generative Adversarial Networks.” ICLR 2018.

---

> > > ### Author Response · Authors · 2023-08-19
> > > **Help check if questions are well addressed.**
> > >
> > > Dear Reviewer  ipyC
> > >
> > > Thanks for your efforts and suggestions again! Could we kindly know if the responses have addressed your concerns and if further explanations or clarifications are needed? Your time and efforts in evaluating our work are appreciated greatly.

---

### Official Review · Reviewer_FCWG · 2023-07-04

**Soundness:** 3 good
**Presentation:** 3 good
**Contribution:** 3 good
**Rating:** 6
**Confidence:** 5

**Summary:**

This paper carefully analyzes the problems of current one-shot video editing and proposes two techniques: Shift restricted Temporal Attention Module (STAM) and Fine-coarse Frame Attention Module (FFAM), to address these issues. Several quantitative and qualitative experiments are conducted to verify the effectiveness of the proposed method.

**Strengths:**

1. The authors meticulously analyze the cause of semantic disparity for video editing and attribute this to the covariate shift of generative feature space brought by newly added temporal attention.
2. To address this problem, the authors propose STAM to alleviate the covariate shift and further improve the temporal consistency with another technique FFAM.
3. The manuscript is well-organized and easy to follow.

**Weaknesses:**

1. Is there any quantitative ablation comparison for the proposed techniques? It would be better to quantitatively analyze these components besides qualitative visual comparison.
2. From the visual ablation study, it seems that the normalized liner mapping contributes to the visual quality a lot. Is there any ablation that only applies normalized linear mapping?

**Questions:**

see weakness

**Limitations:**

The authors have adequately addressed the limitation of the proposed method

---

> ### Author Rebuttal · Authors · 2023-08-09
>
> __Q1__: It would be better to quantitatively analyze components besides qualitative visual comparison.
>
> __A1__: Thanks for pointing out.  We have addressed this by including the quantitative results in the Table 1 of the provided PDF file.
> 1. By replacing the TA module with the proposed STAM (+STAM) in the Tune-A-Video baseline, we observe a noticeable increase in the CLIP score for textual alignment. This demonstrates that STAM effectively alleviates the semantic disparity issue by addressing the covariate shift, thus enhancing the quality of video editing.
> 2.  Substituting the SCA module in baseline with FFAM (+FFAM) results in a boost in the CLIP score for frame consistency. This indicates that FFAM better captures motion information from the given video frames.
> 3. It is worth noting that the CLIP score (_i.e._,  CLIP similarity among frames), which is used to measure frame consistency, is not robust to pose and texture variations and does not accurately reflect temporal consistency. Therefore, we conduct an extra user study: Given 15 pairs of prompts and videos, where object changes are more considered,10 volunteers vote for results with better temporal consistency between the ablated method and baseline. The results indicate that FFAM substantially improves the temporal consistency, while STAM slightly compromises temporal consistency due to the challenge of maintaining motion for more pronounced editing effects. This user study further reinforces  the effectiveness of FFAM to strengthen the temporal consistency.
>
> __Q2__: Is there any ablation that only applies normalized linear mapping?
>
> __A2__: We illustrate the qualitative and quantitative experiments of normalized linear mapping in the provided pdf.
> 1. As shown in Table 1, directly leveraging Normalized Linear Mapping  (the column w/o $IC$) can obviously improve the textual alignment score ($29.02$) compared with baseline model (+FFAM $28.50$). The results also show that $IC$ layer contributes more ($29.51$) to the alleviation, and the effects of two modules can add up to each other ($29.84$).
> 2. As shown in Figure 4, Normalized Linear Mapping is also useful to enhance the visual effects, but the improvement is usually no more than $IC$ layer. And the effects of two modules can add up to each other.
>
> These observations are supported and aligned with our theoretical analysis, where the Normalized Linear Mapping mainly alleviate the variance shift. Hence, in STAM it can be considered as an effective complementary of $IC$ layer.

---

> > ### Comment · Reviewer_FCWG · 2023-08-17
> > **Final decision**
> >
> > Thanks for the authors' response. All my concerns are addressed.

---

> > > ### Author Response · Authors · 2023-08-17
> > > **Thanks for your comments!**
> > >
> > > Dear Reviewer FCWG,
> > >
> > > We are delighted that your concerns have been effectively addressed.  We sincerely appreciate your valuable comments, prompt response, and recognition of our work.

---

### Official Review · Reviewer_8VBg · 2023-07-07

**Soundness:** 2 fair
**Presentation:** 2 fair
**Contribution:** 2 fair
**Rating:** 5
**Confidence:** 4

**Summary:**

This paper claims that existing models suffer from inconsistent video editing results due to covariate shift caused by additional modules for learning temporal information. It proposes a novel model addressing this issue through two modules: Shift-restricted Temporal Attention (STAM) and Fine-coarse Frame Attention (FFAM). STAM replaces Layer Normalization with an Instance Centering layer and utilizes a normalized mapping attention layer to preserve temporal feature distribution and constrain variance shift. FFAM incorporates fine-coarse spatial information to further improve temporal consistency. Experimental results demonstrate the effectiveness of the proposed EI model.

**Strengths:**

1. The paper in detail analysed the issue of covariate shift in temporal consistency, which is a good perspective to cut in, and the proposed method accordingly tackled this issue.

2. Figures 4 and 5 show ablation experiments that illustrate how each component contribute to the final performance.

**Weaknesses:**

1. Figure 3, as well as not few examples in the supplementary video (especially the first half almost), seems to show that the other methods aren't mostly doing well on text information, while having similar/comparable temporal consistency. This seems not to be consistent with the main target and point of the proposed method. In Table 1, the main metric, frame consistency in terms of CLIP score, doesn't show the proposed method's advantange either. I would like to learn more discussions here.

2. The examples in the supplementary videos are better to be listed together with the same input and all other methods, instead of separately one by one. Current layout only has one comparing method per slide, which is not clear and hard to compare with all.

**Questions:**

1. [Minor] The title might be a bit too high-level since this paper just points out and targets at one aspect of the cause in temporal consistency issue. Addressing the specific technical points might be a more proper and solid claim.

**Limitations:**

1. There are many latest related work on this topic emerging recently. Maybe many of them are concurrent work. It is not mandatory, but very good to include more comparisons with some most recent work, in terms of methodology, results and other technical differences, especially for those that have released all code and pretrained models.

---

> ### Author Rebuttal · Authors · 2023-08-09
>
> __Q1__: Results in table1 and Figure 3 are not consistent with the main target and point of the proposed method.
>
> __A1__: In our work, the consistency refers to (L44-46) both semantic alignment of text and edited video, and temporal consistency in cross frames.
> In this context, the results and your observations align with our intended point. Specifically, the observed semantic inconsistency in Tune-A-Video highlights a practical limitation that our approach seeks to overcome. To elaborate:
> 1. Our theoretical analysis associates this issue with covariate shift stemming from newly introduced temporal attention, prompting us to propose STAM as a solution.
> 2. Besides, we introduce FFAM, an efficient enhancement to the prior sparse-casual attention module, which improves frame consistency.
>
> In light of these contributions, the enhanced textual alignment indeed constitutes a part of our broader advancements.
>
> __Q2__: In Table 1, the main metric, frame consistency in terms of CLIP score, doesn't show the proposed method's advantage.
>
> __A2__: Aligned with the objectives of video editing, we aim to enhance semantic alignment of text prompts and edited results, while maintaining temporal consistency. Our proposed EI$^2$ method demonstrates advantageous performance in both these crucial aspects:
>
> 1. Advancement in semantic consistency: Leveraging CLIP score (L287-288) and a comprehensive user study, our method consistently achieves superior results in terms of semantic alignment between edited videos and text prompts. This underscores the effectiveness of our STAM module.
> 2. Advancement in temporal consistency:  Similarly, we conduct the automatic metric, _i.e._, CLIP score for frame consistency   (L285-286) and user study to evaluate the temporal consistency. We would like to clarify the inferior of CLIP score from the distinction between CLIP score and temporal consistency:
>    - **CLIP Score for Frame consistency vs. Temporal Consistency**. It is important to note that the CLIP score primarily evaluates high-level semantic consistency among frames. In contrast, "temporal consistency" traditionally assesses low-level details in cross-frame changes. Thus, the CLIP score may not fully account for artifacts or noise in frames, which are pivotal for visual perception, and can be influenced by multiple factors like content, pose, and texture. This also has been noted in previous works like Dreamix and vid2vid-zero. In this way, while the CLIP score remains a valuable objective evaluation metric, it might not capture temporal nuances accurately.
>
>     To account for this, additional user study is conducted to validate temporal consistency, and demonstrates the improvements of EI$^2$. We depict some examples in the provided PDF file to prove that our method can perform better than TAV in many cases.
>
> In summary, with the effective improvement, EI$^2$ can achieve the better semantic and temporal consistency.
>
> __Q3__: Examples in the supplementary videos is not clear and hard to compare with all.
>
> __A3__: Thanks for pointing out.  We would have wanted to use this layout to provide high-resolution details, since the compressed file format might largely compromise the visual details. In the upcoming revision, we will take steps to enhance the clarity and ensure that the presented way effectively illustrates comparisons.
>
> __Q4__: About title problem.
>
> __A4__: Thanks for pointing out. We will refine this in the revision.
>
>
> __Q5__: It will be better to include more comparisons with some most recent work.
>
> __A5__: We agree that there are indeed many con-current and pre-print works after the submission, and we are actively incorporating these additional references into the revision. We consider that the methods we have currently compared are representatives of this field: the base method for model inflation and tuning (TAV), the extended approach for structure control (Video-P2P), and the method for zero-shot editing (vid2vid-zero). EI$^2$ is aligned with the first category, as it delves into exploring and revealing the way of inflation.

---

> > ### Comment · Reviewer_8VBg · 2023-08-17
> >
> > Thank you for your clear and informative responses:
> >
> > Q1: I acknowledge your explanation and definition of the overall consistency instead of only temporal consistency. Then I think it would be better to more highlight these two aspects respectively in the paper, and especially the title: putting consistency and video together could mostly make readers think mainly about temporal consistency.
> >
> > Q2: I partially agree with your statement. The CLIP score still emphasizes more on semantic aspects and for pure temporal consistency it's not easy to measure. Your method leads mainly in the user study, which is not complex but should be effective. However, I think it's still not solid enough to mainly rely on these metrics. For example, you only show 30 video pairs to 50 volunteers. I think for example Amazon Turk is able to collect 10x or even bigger scale of the survey with not a too big finanacial cost. And it's better to show more videos while it doesn't have to have every video to be viewed by all volunteers. For example, 300 video pairs and 600/900 volunteers, each video pair rated by 2/3 volunteers, should be much more solid (more videos mean less room for cherrypicking) compared to the curret plan, while still acceptable in cost.
> >
> > For video temporal consistency, although there might haven't been a very popular and sound metric, it would still be better to try some more options, such as the simplest differences among frames inside each video (in the form of pixels, SSIM or LPIPS etc.). This can also become part of your contribution.
> >
> > Q3: No problem.
> >
> > Q4: Yes please include your specific techniques etc. in the title instead of describing the problem to solve only. And also please refer to Q1.
> >
> > Q5: No problem.
> >
> > Overall I tend to keep my original ratings, with a kind of slight leaning toward acceptance. The main positive points are the proposed methods and analyses; the main negative points are still about the claims and solid results and metrics.

---

> > > ### Author Response · Authors · 2023-08-17
> > > **Thanks for your comments!**
> > >
> > > Dear Reviewer 8VBg,
> > >
> > > We sincerely appreciate your valuable comments, prompt response, and recognition of the innovative aspects of our work.  We will follow your suggestion to refine the final version.
> > >
> > > Regarding the user study, we agree your point that involving more videos and volunteers would make the results more solid.   But, compared to concurrent works our study **boasts an extensive number** of questionnaires, 1500 in total for 50 participants and 30 videos. In contrast, TAV, Pix2Video (ICCV 2023), and FateZero (ICCV 2023) engaged 5, 37, and 20 participants respectively, in the assessment of 147, 20, and 9 edited videos. We are committed to further expanding our dataset to ensure its robustness and reliability.
> > >
> > > Additionally, we highly value your opinion and the suggested metrics. We are committed to incorporating the relevant experiments into the revised version.
> > >
> > > Once again, we sincerely appreciate your efforts and positive feedback!

---

> > > ### Author Response · Authors · 2023-08-18
> > > **About metrics on video temporal consistency**
> > >
> > > Dear reviewer 8VBg,
> > >
> > > Thanks for your suggestion again. We have tried the SSIM/LPIPS metrics differences among frames inside 75 edited videos.
> > >
> > > |        | Origin |  TAV  | Vid2vid-zero |  EI$^2$ |Video-P2P |  EI$^2$+P2P |
> > > |:------:|:------:|:-----:|:------------:|:-----:|:---------:|:---------:|
> > > | LPIPS↓ |  0.207 | 0.242 |     0.257    |   0.238 | 0.233   | 0.231 |
> > > |  SSIM↑ |  0.692 | 0.667 |     0.644    |  0.672 | 0.677  |  0.678 |
> > >
> > > The origin denotes the metrices on original videos.  It seems that 1) EI$^2$ can achieve better results compared with the baseline Tune-A-Video, indicating the effective improvements. 2) As an orthogonal approach to TAV/EI$^2$, Video-P2P that refines TAV to fit the original video, yields improved metric scores. This advantage can also be combined with EI$^2$. However, it is important to observe that methods incorporating P2P tend to generate artifacts or experience failures in individual frames, impacting their overall human perception quality. We will add this discussion in the revision.

---

### Official Review · Reviewer_eBTE · 2023-07-07

**Soundness:** 3 good
**Presentation:** 3 good
**Contribution:** 3 good
**Rating:** 5
**Confidence:** 5

**Summary:**

The paper provides a comprehensive theoretical analysis of the covariate shift problem in current one-shot fine-tuning methods. In response, the authors propose a novel Shift-restrict Temporal Attention Module, designed to enhance the editing capability. Additionally, they introduce a Fine-coarse Frame Attention Module, which aims to improve temporal consistency. The visual examples presented in the paper demonstrate promising performance.

**Strengths:**

1. The paper offers a comprehensive analysis of the underlying causes of semantic disparity in fine-tuning-based video editing approaches. Through a rigorous examination under the Gaussian assumption, the authors effectively demonstrate the occurrence of covariate shift. Furthermore, the proposed solution put forth in the paper is both logical and well-founded.
2. The paper showcases temporal consistency in the edited results, which is a noteworthy accomplishment.
3. The paper demonstrates a commendable level of clarity and cohesiveness in its presentation, making it easily comprehensible and accessible to readers.

**Weaknesses:**

1. The analysis of covariate shift in this paper relies fundamentally on the assumption of i.i.d. Gaussian distribution. However, given the complex nature of real-world scenarios, this assumption may be overly restrictive. It would be valuable to incorporate a more generalized analysis to account for a broader range of situations.
2. While it is acknowledged that commonly used CLIP-based metrics may not align perfectly with human perception, they remain the most objective evaluation metrics available. Table 1 reveals that the CLIP score for temporal consistency of the proposed method is inferior to that of previous works. Could you please provide further details on the user study, particularly in terms of ensuring sufficient objectivity and randomness?
3. The visual examples depicted in Figure 3 highlight certain deficiencies in the fidelity of the proposed method to the original videos. For instance, the 3rd and 4th frames of the 'Spider-Man' case exhibit noticeable motion and pose disparities compared to the original frames. Additionally, the 4th frame in the 'man talking' case shows an open mouth, deviating from the original video.
4. In addition to assessing temporal consistency and textual alignment, it is crucial to incorporate fidelity evaluation metrics such as PSNR/SSIM/LPIPS as reported in Fatezero and Video-p2p. It would be beneficial to include these metrics in the analysis to provide a more comprehensive assessment of the proposed method.
5. The ablation study only includes one example to demonstrate the effectiveness of the proposed designs, which may not be sufficient. It is recommended to include a quantitative evaluation to strengthen the analysis and provide a more robust assessment of the proposed approach.



**Questions:**

1. In L184, the authors state that randomly initializing the Temporal Attention (TA) module could potentially compromise its effectiveness in terms of editing capability. Considering this, would it be beneficial to explore the possibility of initializing the TA module from well-trained Spatial Attention (SA) modules, as shown useful in other works?
2. Upon reviewing the video comparisons provided in the supplementary materials, it becomes evident that several results from Video-p2p differ significantly from the demonstrations shown on their official website. For instance, the 'Spider-Man' riding case and the 'car driving' case display instances where the inversion step does not appear to have succeeded. I recommend investigating the code and corresponding results to address this discrepancy.
3. A minor suggestion regarding Figure 2: it may be preferable to present the 'Inflation' operation without the surrounding box. Including the box might lead to the misconception that this is a parametric module, which could be misleading for readers.
4. Only style manipulation cases are provided. Could the proposed method successfully address the pose/shape modification and object addition/remocal situations?

**Limitations:**

The authors have discussed the limitations.

---

> ### Author Rebuttal · Authors · 2023-08-09
>
> __Q1__: It would be valuable to generalize analysis to account for a broader range of situations.
>
> __A1__: Thanks for pointing out. We recognize this limitation (Line 335-337), and explore two perspectives for generalization:
>
> 1. Distribution Transition Perspective: In the proof, we approach the covariate shift by analyzing how the quasi-linear structure in the residual-based transformer block affects the distribution. This allows us to extend the analysis to various "good" distributions that exhibit "linearity" and "additivity" (Line 80). Thus, if the input follows a "good" distribution (_e.g._., Gaussian, Gamma, Uniform), according to the proof steps we can also derive a rigorous conclusion for that case.
>
> 2. Moment Change Perspective: Recognizing that moments encapsulate vital distribution information. we emphasize their role in the analysis.  By prioritizing the first two moments (mean and variance) and downplaying higher-order moments, we can align them to mitigate covariate shift. This simplifies STAM's goal from preserving distribution to preserving moments. This perspective applies beyond Gaussian distribution, directly generalizing our analysis to real-world scenarios.
>
> We will add more about these in the revision.
>
>
> __Q2__:  CLIP score for Frame consistency, and user study details.
>
> __A2__:   We would like to clarify the distinction between the "CLIP score for frame consistency" and conventional "temporal consistency (Please refer to A2 for Reviewer 8VBg). We try to align our user study with that of TAV:
> - The edited videos from various methods are presented in a random order. The volunteers are required to select their favorite results that have the best temporal consistency and best aligns with the text prompt in vision.  Since previous works do not release their test prompts, we collect 30 commonly-used prompts in recent works for style, object and background changes.  We recruit 50 volunteers, which are from different regions on the networks. Compared with 5 participants in Tune-A-Video, we consider our survey has better randomness and objectiveness.
>
>
> __Q3__: Deficiencies in the motion fidelity of the method to the original videos.
>
> __A3__: We consider this is a shared challenge across recent works (_e.g._, the "dancing man" in the teaser of TAV),  due to the inflated T2V models :
> 1. inherit the limited editing ability of pre-trained LDM, which struggles with preserving delicate details. Even methods like DreamBooth for frame-by-frame editing encounter challenges in maintaining faithful poses, as observed in experiments with the "surfing man".
> 2. cannot perfectly disentangle motion $m$ and content $c$ of video. When given a text $t=(t_c, t_m)$, TAV/EI$^2$ essentially "generates" the edited video, thus the fidelity of edited pose highly relies on whether the distribution $p(m|t_m)$ or $p(m|t)$ is deterministic.  The inflated model may learn $p(c,m|t) = p(m|c,t) p(c|t)=1$ under the diffusion loss, while it is hard to ensure $p(m|t)=1$, which is true if and only if $m$  and $c$  are totally independent in tuned LDM space.  This means that changes in $t_c$ could lead to fluctuations in the "non-deterministic" distribution $p(m|t)$ or $p(m|c,t)$, affecting $m$ during sampling.
>
> Actually, our results appear natural, with motion closely resembling the originals, evident in surfing and mouth motion in SM video, also new examples in the provided PDF. As explained in L278-281, this validates that EI$^2$'s $p(c|t)$ maintains alignment with the pre-trained LDM, and $p(m|c,t)$ has effectively captured meaningful information.  In contrast, TAV may fail in $p(c|t)$ to produce artifacts or semantic disparity.
>
>
> __Q4__: Fidelity metrics PSNR/SSIM/LPIPS to provide a more comprehensive assessment.
>
> __A4__: Thanks for pointing out. We try to follow Video-P2P to compute LPIPS/SSIM for structure preservation, and PSNR for reconstruction quality. Here the PSNR scale may differ from that in Video-P2P. EI$^2$ can perform better than TAV and Vid2vid-zero in these metrics, and it is also extensible to couple with Video-P2P to enhance the fidelity.
>
> |       | Tune-A-Video | Vid2vid-zero |EI$^2$ |  Video-P2P | EI$^2$+P2P |
> | ----- | ------------ | ------------ | --------- | ------ | ---------- |
> | PSNR↑  | 21.9         | 15.5         | 23.7  | 28.4     | 28.4       |
> | LPIPS↓ | 0.40         | 0.42         | 0.38   | 0.35      | 0.34       |
> | SSIM↑  | 0.58         | 0.57         | 0.60   |0.64      |  0.64       |
>
>
> __Q5__: Quantitative evaluation of the ablation study.
>
> __A5__: Thanks for point out. We have provided this in PDF file. Please refer to A1 for Reviewer k6zc.
>
>
>
> __Q6__: Initializing the TA module from well-trained Spatial Attention (SA) modules.
>
> __A6__: Thanks for pointing out. We try to experiment with this initialization, but it seems to face similar shortcomings. We consider this does not conflict with our analysis of covariate shift.   Additionally, we aspire for the TA module to concentrate more on acquiring temporal information with random initialization, akin to Divided Space-Time Attention in TimeSformer[1], instead of utilizing spatial information as strong guidance.
>
> [1] Bertasius G, Wang H, Torresani L. Is space-time attention all you need for video understanding? ICML 2021.
>
>
> __Q7__: Investigating the code of Video-p2p.
>
> __A7__: After carefully checking, we ensure that we follows their official codes and settings. We think the diverse reproduction is probably caused by different devices and seeds. In actuality, for the sake of impartiality, the outcomes of all comparative approaches are derived from the optimal results under multiple random seeds.
>
>
> __Q8__: Better presentation for Figure 2.
>
> __A8__: Thanks for pointing out.  We will refine this in the revision.
>
>
> __Q9__: Could the proposed method address various situations?
>
> __A9__:  Yes, EI$^2$ can perform well, even better than TAV in these situations. We show some examples in the provided pdf.

---

> > ### Comment · Reviewer_eBTE · 2023-08-12
> >
> > Thanks for your response. Several of my concerns have been addressed. I commend the initiative to involve more volunteers in evaluating temporal consistency. However, it's also essential to include the commonly utilized CLIP-based scores for a comprehensive assessment. Furthermore, based on the visual as well as LPIPS/SSIM outcomes, it's evident that further endeavors to enhance fidelity are warranted.

---

> > > ### Author Response · Authors · 2023-08-12
> > > **Further declaration on CLIP score and fidelity**
> > >
> > > Thank you for your feedback and thoughtful considerations. We appreciate your recognition of our efforts to involve more volunteers.
> > >
> > > - Regarding the CLIP-based scores, we would like to clarify that **they have been provided** in our assessment/ablation to evaluate textual alignment and frame consistency, as highlighted in the tables of the main paper and rebuttal pdf attachment. The results show that EI$^2$ performs considerably on the textual alignment, and comparably on frame consistency (Please refer to A2 for Reviewer 8VBg).
> > >
> > > - Regarding the fidelity, it is a shared challenge across recent works to ensure perfect motion fidelity in edited videos (refer to explanations in A3).  In the case of more pronounced editing effects compared with previous works, EI$^2$  (or +P2P) still showcases **improvements in PSNR/LPIPS/SSIM** metric for structure preservation. In the Figure 1, EI$^2$ achieves **much more faithful motion and identity** compared with previous works.  We believe this demonstrates EI$^2$'s reasonable design.
> > >
> > > We acknowledge the importance of improved fidelity in practice usage, and we are committed to addressing this aspect in our future work. Your insights are valuable to advance our approach.

---

### Official Review · Reviewer_k6zc · 2023-07-11

**Soundness:** 4 excellent
**Presentation:** 4 excellent
**Contribution:** 4 excellent
**Rating:** 7
**Confidence:** 4

**Summary:**

This paper proposes $EI^2$ for the consistent text-driven video editing. To address the semantic disparity problem, it provides a detailed theoretical analysis and proposes Shift-restricted Temporal Attention Module (STAM) to tackle the covariate shift problem introduced by the temporal consistency module. To improve the temporal consistency, it provides a novel Fine-coarse Frame Attention Module (FFAM) to fully exploit the global temporal information while reduce the computational cost. Extensive comparisons with several available video editing methods shows its superiority and the ablation study also illustrates the effectiveness of the STAM and FFAM.

**Strengths:**

- The authors provides a theoretical analysis about the semantic disparity problem in video editing, which makes the paper more convincing.
- Based on the detailed analysis, the authors propose a novel Shift-restricted Temporal Attention Module to tackle the covariate shift, which can imrpove the semantic consistency in the edited video.
- The design of the Fine-coarse Frame Attention Module makes sense, which can balance the performance and the computational cost.
- The visual comparison shows its superiority over existing publicly available methods while the ablation study further illustrates the effectiveness of the proposed method.
- The paper is well-presented.

**Weaknesses:**

- Lack of quantitative results of the ablation study. Though the authors provide the qualitative comparsion result for the abltion study to validate the effectiveness of the STAM and FFAM, it will make the paper more convincing if they provide the some quantitative results for the ablation study.
- Lack of some technical details. For example, when conducting the temporal attention, how many frames does the model take into consideration?

**Questions:**

- The duration of the generated video is quite short (about 1s-2s). How does the proposed methods perform when generating long video? (upto 5s-10s)
- Will the trianing and inference code be released?

**Limitations:**

The authors have discussed the limitations in the main paper. More potential limirations please refer to Questions.

---

> ### Author Rebuttal · Authors · 2023-08-09
>
> __Q1__: Lack of quantitative ablation study to validate the effectiveness of the STAM and FFAM.
>
> __A1__: Thanks for pointing out.  We have addressed this by including the quantitative results in the Table 1 of the provided PDF file.
> 1. By replacing the TA module with the proposed STAM (+STAM) in the Tune-A-Video baseline, we observe a noticeable increase in the CLIP score for textual alignment. This demonstrates that STAM effectively alleviates the semantic disparity issue by addressing the covariate shift, thus enhancing the quality of video editing.
> 2.  Substituting the SCA module in baseline with FFAM (+FFAM) results in a boost in the CLIP score for frame consistency. This indicates that FFAM better captures motion information from the given video frames.
> 3. It is worth noting that the CLIP score (_i.e._,  CLIP similarity among frames), which is used to measure frame consistency, is not robust to pose and texture variations and does not accurately reflect temporal consistency. Therefore, we conduct an extra user study: Given 15 pairs of prompts and videos, where object changes are more considered, 10 volunteers vote for results with better temporal consistency between the ablated method and baseline. The results indicate that FFAM substantially improves the temporal consistency, while STAM slightly compromises temporal consistency due to the challenge of maintaining motion for more pronounced editing effects. This user study further reinforces  the effectiveness of FFAM to strengthen the temporal consistency.
>
> We will add these in the revision.
>
>
> __Q2__: Lack of some technical details, _e.g._, how many frames does the model take into consideration?
>
> __A2__: Here we provide more details to make the technology clearer:
>
> 1. Model details:  EI$^2$ inflates TTI to TTV model with the integration of STAM and FFAM modules, for an intuitive comprehension of the technique, we report details of these two modules:
>     a) FFAM strictly aligns the illustration in Eq. (10) and Figure 2, which half down-samples features of all other frames by the bilinear interpolation, thus does not introduce new parameters.
>     b) In STAM, the frame length is the same as the frames of video. Power iteration algorithm is used to fast approximate the max singular value, thus the spectral normalization does not bring extra time cost. For avoiding the illness of small singular value on normalization to magnify variance, we only normalize the matrix with spectral norm larger than 1 in practice.
> 2. Training details: We perform the training at resolution of 512 × 512. Following Tune-A-Video, we only fine-tune unfrozen parameters including query weight $W_q$ of FFAM and cross attention, and all weights in STAM. The optimization runs 500 steps on a learning rate 3e−5 (batch size = 1) with AdamW optimizer.
> 3. Inference details: We first use DDIM inversion to get the latent code the video, then the edited results are obtained via the DDIM sampling with given prompt after 50 steps. Thanks to STAM, we can set the classifier-free guidance scale to a small value, while Tune-A-Video works with a large value. In our paper, it is set to 7.5 same as the default value of LDM.
>
> __Q3__: How does the proposed methods perform when generating long video?
>
> __A3__: EI$^2$ demonstrates impressive results for long videos. We have included examples of editing for a long video sequence (about 8s) in the attached rebuttal PDF, where the video is sampled with a sample rate 2, as same as the setting in Tune-A-Video, to get 36 frames in totally. These examples underscore the capability of EI^2 to maintain its semantic and temporal consistency when applied to longer video sequences and various tasks (_e.g._, object, style, background changes), surpassing the baseline Tune-A-Video.  We will provide more examples in the revision.
>
> __Q4__: Will the training and inference code be released?
>
> __A4__: Yes, we will make the code publicly available.

---

> > ### Comment · Reviewer_k6zc · 2023-08-18
> > **Post-rebuttal Comment**
> >
> > Thanks for the response. All  of my concerns have been addressed. I will keep my origianl rating.

---

> > > ### Author Response · Authors · 2023-08-18
> > > **Thanks for your comments!**
> > >
> > > Dear Reviewer k6zc,
> > >
> > > We are delighted that your concerns have been thoroughly addressed. We sincerely appreciate your valuable comments, prompt response, and recognition of our work.

---

### Author Rebuttal · Authors · 2023-08-09

We thank all the reviewers for their thoughtful reviews and insightful comments on our work!  It is encouraging to hear the feedback from reviewers that:

1. The paper is commendably easy to follow. It is well-written and presented in terms of organization, logic, commendable clarity and cohesiveness (k6zc,FCWG,eBTE).
2. The theoretical analysis is insightful. It elaborates the covariate shift problem to cause semantic inconsistency in video editing with a meticulous, clear and rigorous manner. (k6zc, FCWG, eBTE, 8VBg, ipyC).
3. On this basis, the proposed EI$^2$ is novel and well-grounded, where STAM is well-founded to alleviate this semantic inconsistency (k6zc, FCWG, eBTE, 8VBg), and the way of FFAM to improve the temporal consistency while balancing the computational cost also makes sense (k6zc,FCWG,ipyC).
4. The experiments including comparison and ablation show promising performance and superiority over existing publicly available methods (k6zc,eBTE,8VBg,ipyC).

Moreover, we appreciate and highly value the questions raised by the reviewers for their key guidance to consolidate this work. We have taken these questions seriously, and provided additional experiments and clarifications in responses below.  We hope these responses could address reviewers' concerns.

---

### Decision · Program_Chairs · 2023-09-21

**Decision:**

Accept (poster)

**Comment:**

This paper introduces the EI$^2$ model to enhance the consistency of Text-to-Image (TTI) diffusion models. The authors identify that added modules in state-of-the-art TTI methods,  such as temporal layers, induce a covariate shift that negatively impacts video edition. They offer a theoretical analysis of the covariate shift and introduce specific modules to mitigate this effect. These modules include the Temporal Attention Module (STAM) and the Fine-coarse Frame Attention Module (FFAM).

The paper received a predominantly positive evaluation, with four reviewers recommending acceptance and one recommending rejection. The main concerns raised by reviewers related to assumptions made in the theoretical analysis, the scale of the user study, and the need for clarifications in experiments, including aspects such as the quality of video editing results, chosen metrics, and potential conflicts between FFAM and STAM. The rebuttal adequately addressed these concerns, and the reviewers maintained their initial recommendations after the rebuttal. Reviewer RipyC maintains a borderline reject recommendation primarily due to concerns about the degraded temporal consistency of the proposed approach.

The AC has thoroughly reviewed the submission and the discussions. The AC considers that the paper offers an interesting analysis of covariate shift in video editing, presenting both theoretical results and relevant solutions to overcome it. They believe that the evidence on temporal consistency could have been strengthened by including comparisons to Neural Layered Atlases baselines such as [A], as suggested in the rebuttal. The FFAM and STAM components seem to introduce a trade-off between consistency and semantic alignment, but the AC considers that the provided results are overall convincing. Therefore, the AC recommends paper acceptance but highly encourages the authors to take into account reviewers' feedback when preparing the final version.

[A] Text2LIVE: Text-Driven Layered Image and Video Editing. O. Bar-Tal et al., ECCV 2022.